# Chalcopyrite Leaching in the Presence of Isopropanol—The Kinetic and Mechanistic Studies

**DOI:** 10.3390/ma17040824

**Published:** 2024-02-08

**Authors:** Tomasz Michałek, Krzysztof Pacławski, Krzysztof Fitzner

**Affiliations:** Faculty of Non-Ferrous Metals, AGH University of Krakow, Mickiewicza Ave. 30, 30-059 Krakow, Poland; tomaszm@agh.edu.pl (T.M.); paclaw@agh.edu.pl (K.P.)

**Keywords:** copper leaching, chalcopyrite, kinetics, batch reactor

## Abstract

Oxidative leaching, as a basic step of the hydrometallurgical process of pure copper production from chalcopyrite, is a slow process in which mineral acids with strong oxidants addition are usually used as a leaching medium. It was found experimentally that the copper leaching from chalcopyrite in the H_2_SO_4_–H_2_O_2_–H_2_O system, in the presence of isopropanol (IPA) and under other conditions (H_2_O_2_ concentration, rate of mixing and temperature), takes place with satisfactory rate and efficiency. To quantify how much the change of these crucial variables affects the rate of the process, experimentally obtained kinetic curves (conversion over time) were analyzed using a Shrinking Core Model (SCM). The determined values of the copper leaching rate constants (k_obs_) confirmed the positive influence of increasing IPA and H_2_O_2_ concentrations as well as the temperature on the kinetics and efficiency of the leaching. The kinetic studies were also supported by using X-ray diffraction (XRD), ^57^Fe Mössbauer spectroscopy, scanning electron microscopy (SEM), and adsorption measurements. The positive influence of IPA was explained by its stabilizing role for iron compounds (hematite, magnetite, and pyrite), which are catalysts during the Cu dissolution, as well as H_2_O_2_ protection from decomposition during free radical reactions. Finally, the optimal conditions for efficient leaching, the rate-limiting step as well as the mechanism suggestion of the copper dissolution, were given.

## 1. Introduction

Leaching, which is a method leading to the transfer of metallic elements from minerals to the liquid phase, can be conducted by a variety of ways [1]. This important technological step determines further processing conditions, especially the methods (e.g., extraction, sorption, chemical reduction, electro-reduction, precipitation, etc. [2,3,4,5]) and the external conditions of pure metal production. Depending on the form of the metal (metallic or metal compound), the proper type of leaching system (oxidative or non-oxidative one [6]) with specific chemical composition must be applied. The most popular and efficient agents for copper leaching from chalcopyrite are aqueous solutions containing either mineral acids with ferric ions [7,8,9] or hydrogen peroxide addition [10,11,12]. Generally, the equation describing the chalcopyrite leaching reaction in acidic media with the addition of hydrogen peroxide can be written as follows:2CuFeS_2_ (s) + 17H_2_O_2_ + 2H^+^ = 2Cu^2+^ + 2Fe^3+^ + SO_4_^2−^ + 18H_2_O(1)

While chalcopyrite leaching has been studied for a long time under different conditions (up to now, in the world market, ca. 80% sources for pure copper production are still chalcopyrite), due to economic reasons, new methods of process improvement are constantly in demand. From a practical point of view, two of them seem to be essential: photocorrosion (using the semiconductive nature of chalcopyrite [13]) and leaching in acidic media with different additives [14]. While the former has not yet been studied in detail (at least the kinetics of the photocatalytic leaching), the latter is already partly proven (e.g., by Solís-Marcíal et al. [14,15,16] or by Ghomi et al. [17]) to be effective for improving the process of leaching. These two scientific groups suggested benefits of non-polar organic solvents present in the leaching solution playing the role of a stabilizing factor of the cuprous ions (the product of chalcopyrite leaching) or stabilizing effect for oxidant (H_2_O_2_), respectively. These phenomena resulted in heightened process efficiency and had a positive impact on the reaction rate.

In our studies, we determined quantitatively how the rate of the copper leaching from chalcopyrite is affected by varied factors, i.e., the presence of isopropanol (IPA) in the system, the rate of solution stirring (*v_SR_*), H_2_O_2_ concentration (C_H2O2_), and the temperature (T). As an integral part of kinetic studies, the microscopic (SEM) observations of the chalcopyrite morphology supplemented by BET isotherm determination and the ^57^Fe Mössbauer analyses of the phase composition were performed. The obtained experimental results gave the basis for the suggestion of optimal process conditions and the possible mechanism of the copper leaching from chalcopyrite.

## 2. Materials and Methods

The starting material, which was used to prepare the powder samples for copper leaching experiments, were natural chalcopyrite nuggets (San Martin Mine, Zacatecas, Mexico). They were pre-crushed in a mortar and then ground for five minutes using a roller-ring mill. As a result, a fine powder fraction of the mineral, with particles from 5 to 100 µm in size, was obtained (Figure 1). All microscopic analyses of samples surface morphology were performed using Scanning Electron Microscopy (SEM) (HITACHI, SU-70, Tokyo, Japan) with an accelerating voltage of 10 kV. Prior to SEM analysis, the samples were sputter-coated with gold in an argon atmosphere (99.999%, Air Liquide, Kraków, Poland) using the DC magnetron sputter (Emitech, K575XD, Fall River, MA, USA).

The chemical and phase compositions of the powder samples were analyzed using combined methods of X-ray Fluorescence XRF (RIGAKU, Miniflex, Tokyo, Japan) with a palladium lamp as a source of radiation, X-ray Diffraction XRD (RIGAKU, Primini, Japan) with the use of a copper tube (λ = 1.54059 Å) and ^57^Fe Mössbauer spectroscopy (RENON, MS-4, Kraków, Poland). Additionally, for the carbon content determination, an exhaust gas (CO and CO_2_) analysis of burnt sample was performed using the infra-red IR method (LECO, CS-600, St. Joseph, MI, USA). During this measurement process, the chalcopyrite sample (0.1 g), mixed with LECOCEL II HP and Iron Chip HP fluxes (each weighing 1 g), undergoes inductive heating in high-purity oxygen (99.999%, Air Liquide, Poland). This process leads to combustion, converting carbon into oxides. Gas content is then analyzed after passing through a catalytic bed and filters.

The experiments of chalcopyrite leaching were conducted in a stationary glass reactor (600 mL capacity) with a cover, thermostated and equipped with a magnetic stirrer. The scheme of this experimental arrangement is shown in Figure 2. The entire system was shielded from external light to ensure that the kinetics would not be influenced by the semiconductive nature of chalcopyrite [13].

In each experimental run, a sample of chalcopyrite powder with a mass of 2.0 g was placed in the reactor, which was filled with 200 mL of leaching solution. The solution consisted of 0.5 M sulfuric acid (30%, pure p.a., Chempur, Piekary Śląskie, Poland), 0–2.0 M isopropanol (60.10 g/mol, pure p.a., Chempur, Piekary Śląskie, Poland), and 0–2.0 M hydrogen peroxide (30%, pure p.a., POCH, Gliwice, Poland), depending on the variant of the experiment. During the leaching time (after 20, 40, 60, 90, 120, and 180 min), 2.0 mL sample for copper ions content analysis was taken. The analysis was performed using Microwave Plasma–Atomic Emission Spectroscopy (AGILENT, 4210 MP AES, Santa Clara, CA, USA). In this way, the kinetic curves (the conversion, X_Cu_, over time) were obtained. The conversion X_Cu_ was defined as a ratio of Cu content in the solution to Cu initial content in the solid sample of the chalcopyrite. Collected data were subjected to mathematical analysis to fit them with an appropriate kinetic model, and to determine the values of the Cu leaching rate constants (k_obs_). For kinetic data description, the Shrinking Core Model (SCM) [18] has been applied in three variants, describing different rate-limiting steps:Product diffusion through the film: X_Cu_ = k_obs_t (2)
Product diffusion through the ash: X_Cu_ + (1 − X_Cu_)ln (1 − X_Cu_) = k_obs_t (3)
The reaction: 1 − (1 − X_Cu_)^1/2^ = k_obs_t(4)

The k_obs_ values corresponded to the slope of linear dependencies obtained during the fitting procedure. The results obtained from kinetic measurements assisted us in determining the optimal leaching conditions and identifying the rate-limiting step of the studied reaction. Since each limiting step is characterized by a different equation, by mathematically applying these equations to experimental data, the step with the highest coefficient of determination (*R*^2^) is considered the limiting step. SCM assumes constant particle geometry, with different shapes of particles described by distinct equations. In the case of this study, the best fit was achieved for the cylinder-shaped particles, and Equations (2)–(4) correspond to this specific geometry.

After leaching experiments performed in the presence of IPA, the morphology and the phase composition of solid samples residue were analyzed using SEM and the ^57^Fe Mössbauer method, respectively. The registered kinetic curves, the results of their fitting, and the results discussion are presented in the next subsections.

The samples were also subjected to BET analyses (Micromeritics ASAP 2010, USA), after their preliminary degassing process at 50 °C for a duration of 24 h. The initial data collection commenced at a pressure of 9–10 mmHg. The measuring cell temperature was maintained at 77.35 K, and each sample was examined at 10 min intervals.

## 3. Results

The results of XRF (for metallic elements) and IR (for carbon) chemical analysis of the chalcopyrite used in all kinetic experiments are shown in Table 1.

The chemical composition of the sample before leaching shows that except for main elements (Cu, Fe, and S), a slight amount of Ag, Pb, Zn, Ni, Mn, and C are present. The results also indicated a greater Fe and S content than would appear from the CuFeS_2_ stoichiometry (pure CuFeS_2_ should contain approximately 34.5% Cu, 30.5% Fe, and 35% S [19]). This fact can be explained by the phase analysis results indicating that magnetite (Fe_3_O_4_) and pyrite (FeS_2_) (Figure 3a,b) are also present in the material. Detailed parameters of phases from ^57^Fe Mössbauer analysis, as well as the names of the identified phases, are presented in Table 2. The identification was conducted by comparing the resulting values to those found in the literature.

### 3.1. Effect of Isopropanol

To analyze the influence of isopropanol (IPA) addition on the kinetics of copper leaching from chalcopyrite, experiments in the presence of 0, 1.0, and 2.0 M of IPA in the reacting system were conducted. Experimental conditions are given in Table 3.

The obtained kinetic curves and the corresponding fits of kinetic equation (according to Equation (3)) are shown in Figure 4a,b, respectively.

The analysis of these results clearly shows the beneficial influence of the IPA on the kinetics of copper leaching in H_2_SO_4_–H_2_O_2_ solution. It is clearly seen from Figure 4b that the increase in the slope (corresponding to k_obs_ of the reaction) of the relationships takes place with the increase in IPA content in the system. The k_obs_ value increases twice, from 4 × 10^−4^ min^−1^ (for the solution without IPA presence) to 8 × 10^−4^ min^−1^ (for the solution containing 2 M of IPA). The same beneficial effect of IPA addition can be observed in Figure 4a for leaching efficiency, represented by maximum conversion value (X_Cu,max_) reached after 180 min. There is a visible difference in conversion (X_Cu,max_) between a run without IPA and that with the presence of 1.0 M IPA in the system. The X_Cu,max_ values are equal to 35% and 45%, respectively. It can also be noticed that further increase in IPA concentration (up to 2.0 M) did not have a significant impact on the conversion (X_Cu,max_ = 48%) in the analyzed time range. However, since the 2.0 M value has proven to be the most effective among tested concentrations, further kinetic experiments were performed at this concentration.

For evaluation of isopropanol influence on the change of surface morphology and phase composition in leached chalcopyrite samples, SEM (Figure 5a,d) and ^57^Fe Mössbauer (Figure 6, Table 4) analyses were performed. Additionally, as a supplementary study, the change of the substrate (chalcopyrite) surface was estimated from the measurements of N_2_ adsorption using BET analysis (Figure 7a,b).

The SEM images vividly depict the influence of IPA on chalcopyrite surface morphology. The un-leached sample (Figure 5a), characterized by a relatively undeveloped surface, and the leached chalcopyrite sample without IPA addition (Figure 5b) exhibit similarities. However, following the addition of IPA (1.0 and 2.0 M IPA), notable transformations unfold—the chalcopyrite surface becomes subtly smoother, spongy, and more porous, as illustrated in Figure 5c,d.

The progress of surface expansion was also experimentally confirmed from the adsorption studies. The comparison of BET isotherm data for the un-leached sample (Figure 7a) and samples leached with the addition of IPA (Figure 7c,d) showed a slight increase in the surface area from 0.483 m^2^/g to 0.537 and 0.498 m^2^/g for concentrations of IPA of 1 M and 2 M, respectively. While this increase may be subtle, the analysis of the surface of the sample leached without any IPA present in the system (Figure 7b) shows that its surface area decreased substantially when compared to chalcopyrite before leaching (to 0.354 m^2^/g). Such a decrease is not a desired effect during the leaching process, as it may result in a more difficult penetration of the leaching solution to the unreacted core of solid particles. Summarizing this part of our studies, it appears that the addition of IPA into the reacting system leads to improved surface development. As a result, the enhancement of copper oxidization is demonstrated by the rate increase in the leaching process (the k_obs_ value increases twice when comparing leaching without IPA and with 2 M IPA). However, as an IPA concentration of 1 M resulted in a more developed surface than a concentration of 2 M, while simultaneously achieving a lower k_obs_ value, it is safe to say that this mechanism is not the only way by which the presence of IPA influences this process.

From the ^57^Fe Mössbauer spectra analysis (Figure 6a–c), one can observe an increasing contribution of the iron compounds (Fe_2_O_3_, Fe_3_O_4_, and FeS_2_) in solid residue when the C_IPA_ increases in the leaching medium. These compounds are less leachable and remain relatively stable during the leaching process. It is highly likely that these compounds, stabilized by the IPA on the reaction surface, play the role of catalysts during the copper dissolution. The mechanism of catalytic activity of ferric ions was already suggested by Hirato et al. [8]. It is also worth mentioning the stabilizing role of IPA for hydrogen peroxide, which was described by Ghomi et al. [17]. The authors concluded that alcohol protects H_2_O_2_ from decomposition in a free-radical reaction. To summarize, both phenomena mentioned give the synergetic kinetic effect resulting in the acceleration of the copper dissolution from chalcopyrite samples. Considering the obtained results, further optimization of the leaching conditions was conducted under constant C_IPA_ = 2.0 M applying different rates of stirring (*v_SR_*), H_2_O_2_ concentrations (C_H2O2_), and temperatures (T). These results are described in the next sections.

### 3.2. Effect of Stirring Rate

Due to the heterogeneous character of the studied reaction, the diffusion (either a transfer of an oxidant to the chalcopyrite surface or a product to the bulk solution) under certain conditions can play a role of the kinetic limiting factor of the overall process. Thus, it was expected that the rate of stirring (*v_SR_*) effect should give an answer to the question of whether the copper leaching process is diffusion- or reaction-controlled under the studied conditions. To describe the effect quantitatively, kinetic experiments at different *v_SR_* values (400, 700, and 1000 RPM) were conducted. Experimental conditions are given in Table 5.

The analysis of experimental kinetic curves (Figure 8) shows that increasing rate of stirring has a negative impact on the leaching rate. The leaching rate constant was inversely proportional to the rate of mixing. The k_obs_ value changes from 8 × 10^−4^ to 5 × 10^−4^ min^−1^ when the *v_SR_* increases from 400 to 1000 RPM, respectively. This relationship contradicts the usually expected dependence according to which the intensive reactants stirring results in their better contact and consequently to acceleration of the leaching reaction. The observed opposite behavior of the reacting system can be explained by faster decomposition of hydrogen peroxide (the oxidant of metallic copper) under conditions of its relatively high concentration (2.0 M H_2_O_2_). Indeed, an intense gas (O_2_) evolution and solution foaming was noticed. The effect of the reaction inhibition was also reported by Sokić et al. [12] and Gok et al. [27], who conducted copper leaching in aqueous solutions of mineral acids (H_2_SO_4_ and HNO_3_, respectively) with a large excess of H_2_O_2_. Thus, it can be concluded that too intensive mixing results in the decrease in a copper oxidant concentration in the leaching solution, and finally it hinders copper leaching kinetics. Considering the experimental results, it was assumed the stirring rate of 400 RPM was the optimal one among the tested rates, and then it was applied in further studies.

### 3.3. Effect of H_2_O_2_ Concentration

Hydrogen peroxide, as an oxidant, plays a crucial role in the copper leaching from chalcopyrite. However, it is clear from our previous experiments that H_2_O_2_ in contact with a mineral, especially when its surface is well developed, continuously decomposes. This fact underscores the challenge of selecting an appropriate initial concentration of H_2_O_2_ in the system, as it significantly impacts the final reaction yield. In order to find the optimal H_2_O_2_ concentration for the copper leaching, kinetic experiments at different C_H2O2_ values (0, 0.5, 1.0, and 2.0 M) were carried out. Experimental conditions are given in Table 6.

The obtained kinetic curves (Figure 9) clearly show the beneficial effect of H_2_O_2_ on the copper leaching efficiency, as well as the kinetics. Without its presence in the system, the final conversion (X_max_), registered after 3 h of the leaching process, was merely 2.3%. Application of 0.5, 1.0, and 2.0 M H_2_O_2_ results in significant increase in the conversion value up to 37, 48, and 54%, respectively. Also, the determined rate constants (k_obs_) clearly reflect the positive impact of C_H2O2_ increase. The change of C_H2O2_ from 0 to 2.0 M yields a k_obs_ increase of three orders of magnitude (from 2 × 10^−6^ to 1.2 × 10^−3^ min^−1^). However, despite the greatest positive impact on the leaching kinetics being from high (2.0 M) H_2_O_2_ concentration, intensive decomposition of H_2_O_2_ was observed during the process. Due to the exothermic nature of this decomposition, the reactor required cooling during the first 30 min of the experiment. Taking into account the synergy of these phenomena, we decided to use a C_H2O2_ concentration of 1.0 M in experiments conducted at different temperatures (Section 3.4).

### 3.4. Effect of Temperature

To determine the temperature influence on the rate constant k_obs_ and on the conversion value X_max_, as well as to estimate the enthalpy and entropy of activation for the studied reaction, the kinetic curves at different temperatures (20, 30, 40, and 50 °C) were registered. Experimental conditions are given in Table 7.

The obtained graphical relationships of conversions over time and the corresponding fits of kinetic equation fits are presented in Figure 10a,b, respectively.

From the data shown in Figure 10, the effect of increasing temperature from 20 to 50 °C is evident. The 9-fold increase in k_obs_ value from 2 × 10^−4^ to 18 × 10^−4^ min^−1^, as well as the change in conversion value (X_max_) from 24 to 70%, are derived. It is worth noting that under 50 °C, dynamic oxidant decomposition is observed. Thus, it can be suggested that the optimal leaching temperature should be lower than 50 °C.

Using the linear form of Eyring [28] and Arrhenius’ [29] equation, from the slope and intercept of corresponding graphs (Figure 11a,b), the activation parameters were determined. The obtained values of the enthalpy (ΔH^#^) and the entropy (ΔS^#^) of activation in Eyring’s equation as well as of activation energy (*E_A_*) and preexponential factor (A) in Arrhenius’ equation, are presented in Table 8.

## 4. Discussion

For a large excess of copper oxidant (H_2_O_2_) and Cu^2+^ counter ions (SO_4_^2−^) in the solution (as it was applied in the experiments), the rate of copper ions appearance in the system under reaction-controlled conditions can be generally described as a differential first-order kinetic equation:dC_Cu2+_/dt = k_obs_∙f ([Cu])(5)
where [Cu] is the copper content in the solid chalcopyrite sample.

However, the integral form of Equation (5) is represented by Equation (3) under the assumption of cylindrical shape of particles. This Equation has worked best for the kinetic results interpretation (Figure 4b, Figure 8b, Figure 9b and Figure 10b) and considered the diffusion phenomenon. The data fitting with *R*^2^ from 0.96 to 0.99 for kinetic curves obtained at different IPA concentrations (C_IPA_), stirring rates (*v_SR_*), hydrogen peroxide concentrations (C_H2O2_), and temperatures (*T*) suggests that the rate-limiting step of the process is the transport of dissolution product (Cu^2+^ ad–ions) through the ash layer. It is worth mentioning that for a description of chalcopyrite leaching kinetic data, this SCM model was commonly used by several authors [10,11,14]. Thus, it was assumed that this step is rate-limiting for the reaction of copper leaching from chalcopyrite. The same model was used also by Sokić et al. [12] and Gok et al. [27]. These authors conducted leaching experiments in similar systems (H_2_SO_4_–H_2_O_2_ and H_2_SO_4_–HNO_3_, respectively), but without IPA presence in both cases. Due to the close similarity between the leaching conditions in our studies and those applied by Sokić et al. [12] (with identical 1 M H_2_O_2_ concentrations and the same limiting step in the SCM model), we compared the activation energy values determined in both cases. The lower E_A_ value (approximately 25% lower) determined in our studies indicates more favorable conditions (with IPA additive) for copper leaching in such a system.

The role of various alcohols, along with some other polar organic reagents such as ethylene glycol, acetone, and formic and acetic acids, has already been explored in the literature, as seen in the work of Ruiz-Sánchez et al. [30]. According to these authors, short-chain organic additives inhibit the hydrogen peroxide dismutation reaction and slow down its decomposition in the leaching solution. Although hydrogen peroxide is relatively stable in acidic solutions, the presence of copper and iron promotes its decomposition, generating HO* radicals through Fenton and Fenton-type reactions [31]. Among the organic additives tested by Ruiz-Sánchez et al. [30], cyclohexanol proved to be the most effective due to its resistance to mineralization by HO* radicals, resulting in the highest percentage of leached copper. Following cyclohexanol, alcohols (methanol, ethanol, and isopropanol) were the next best additives, showing similar copper conversion rates (50–51%), with isopropanol attaining the lowest mineralization (8.5% as compared to 98% for methanol). As a result, IPA emerges as a promising additive in certain leaching systems, bolstered by its low cost and the possibility of sustainable production from biomass [32]. In the studied reacting system, the positive role of IPA is evident. It manifests itself via the increase in the rate constant (k_obs_) of the copper dissolution reaction as well as by an increase in conversion efficiency (X_max_) of the reaction (Section 3.1).

It is worth mentioning the role of IPA in the behavior of the iron compounds present in the leached material. From the comparison of the ^57^Fe Mössbauer spectrograms (Figure 6), it is evident that the leaching in the presence of increasing amount of IPA results in reduced dissolution of these compounds from chalcopyrite samples. This effect was also presented in a work of Solis-Marcial et al. [16], where methanol was tested. They suggest that the presence of alcohol stabilizes cuprous ion (Cu^+^), which then partakes in the redox reaction, resulting in oxidation of copper and simultaneous reduction of iron. Considering the literature data [7,8,9] indicating that ferric ions can be a catalyst for Cu oxidation, such a mechanism in the studied system also cannot be excluded. The Fe^3+^ ions coming from magnetite, as well as from Fe^2+^ oxidation, may accelerate the copper dissolution while undergoing reduction themselves. Summarizing, in addition to the stabilizing role of IPA in preventing H_2_O_2_ decomposition in the system, alcohol protects the iron oxides and sulfides from fast dissolution. Consequently, they can further play a catalytic role in the copper dissolution process.

To refer to the literature data presented by other authors, the comparison of experimentally determined *E_A_* value, the leaching medium composition, and the SCM kinetic model applied for kinetic data description was performed (Table 9).

One can tell from these data that the E_A_ determined in our studies corresponds well to the values obtained by other authors conducting experiments in solutions containing both H_2_SO_4_ and H_2_O_2_. It can also be seen from the data shown in Table 9 that Adebayo et al. [10] reported an *E_A_* value of only 39 kJ/mol, which is significantly smaller than that given this study. They used only sulfuric acid and hydrogen peroxide as a leaching medium (without any alcohols), although the hydrogen peroxide was significantly more concentrated in their work. Another case of a system with only sulfuric acid and hydrogen peroxide as a leaching medium is the study of Sokić et al. [12]. They used a higher concentration of sulfuric acid than Adebayo et al. [10] while using a lower concentration of hydrogen peroxide. This resulted in an *E_A_* value of 80 kJ/mol, which may suggest that increasing the H_2_O_2_ concentration in the leaching medium may decrease the *E_A_* value of the process (as it is very unlikely that lower sulfuric acid concentrations caused those differences in *E_A_* values). However, the dynamic decomposition of hydrogen peroxide is observed when its concentration is relatively high (see Section 3.3). H_2_O_2_ decomposition is also observed at elevated temperature (50 °C) (see Section 3.4). Both facts tend to preclude leaching conditions with high concentrations of C_H2O2_ and T in practice. Only the application of the leaching medium with optimal composition (C_H2SO4_ = 0.5 M, C_IPA_ = 2.0 M, C_H2O2_ = 1.0 M) and temperature (T = 40 °C) determined in this work offers the chance for stable conduction of the leaching process. However, applying these conditions, satisfactory process efficiency can be achieved in a time longer than 3 h.

As the used rate equation suggested, the transfer of copper ad–ions from the chalcopyrite surface to the solution is controlled by diffusion of Cu^2+^ ions through the ash layer. This suggestion indicates the significant role of stirring rate (*v_SR_*) as well as the role of the reaction surface, which both ensure the effective penetration by the leaching medium and easy transport of the products. In the studied heterogeneous reaction, specific due to the sensitivity of the oxidant (H_2_O_2_) to too high a *v_SR_* value (see Section 3.2), its concentration (see Section 3.3), and the temperature (see Section 3.4), these variables were optimized and can be partly controllable.

## 5. Conclusions

Summarizing the results of this work, the following conclusions can be drawn:The presence of isopropanol (IPA) in the leaching system enhances the kinetics of copper dissolution from chalcopyrite. The observed leaching rate constant (k_obs_) changes by as much as twice (from 4 × 10^−4^ to 8 × 10^−4^ min^−1^) when the IPA concentration (C_IPA_) increases from 0 to 2.0 M, respectively. The IPA also improved the efficiency of the studied reaction. The final conversion (X_max_) of the copper leaching conducted without IPA in reacting system reaches 35%, while the application of 1.0 and 2.0 M of IPA results in increases in X_max_ to 45 and 48%, respectively.The iron compounds (Fe_2_O_3_, Fe_3_O_4_ and FeS_2_) present in the chalcopyrite are more stable (less leachable) during the leaching process when IPA is added into the solution. Under such conditions, the enhancement of the copper leaching efficiency as well as the rate of the process are evident. It is probably caused by the synergy of two phenomena: the catalytic impact of iron compounds on the rate of Cu dissolution and the stabilizing role of IPA for H_2_O_2_.The microscopic and BET isotherm analyses prove that the IPA presence affects the surface morphology of the leached chalcopyrite. Under such conditions, the spongy and enlarged substrate surface facilitates improved contact with the leaching medium, allowing for easier penetration into the unreacted core of solid particles. This, in turn, promotes a more efficient leaching process.The increase in the stirring rate (*v_SR_*) causes a negative effect on the copper leaching kinetics. In the range from 400 to 1000 RPM, the rate constant (k_obs_) decreases from 8 × 10^−4^ to 5 × 10^−4^ min^−1^. The same relation refers to the conversion (X_max_), which decreases from 48 to 39%, respectively. The leaching inhibition with the *v_SR_* increase is explained by the better reactants contact at higher *v_SR_*, which results in a rapid decrease in oxidant (H_2_O_2_) concentration due to its faster decomposition on the enlarged substrate surface.The application of the optimal H_2_O_2_ concentration is required during the leaching. The lack of H_2_O_2_ presence in the system results in very slow reaction (k_obs_ = 0.2 min^−1^) and a very small final conversion (X_max_ equal to only 2.3%). The increase in H_2_O_2_ concentrations from 0.5 to 2 M results in significant increases in both k_obs_ (from 5 × 10^−4^ to 12 × 10^−4^ min^−1^, respectively) and X_max_ (from 37 to 54%, respectively). However, using concentration C_H2O2_ = 2.0 M in the leaching medium results in a foaming of the reacting solution due to the H_2_O_2_ decomposition with corresponding O_2_ release.The temperature significantly influenced the rate and the efficiency of the process. In the temperature range from 20 to 50 °C, a 9–fold increase in k_obs_ value (from 2 × 10^−4^ to 18 × 10^−4^ min^−1^) as well as a ca. 3–fold change of conversion value (X_max_) (from 24 to 70%) is observed.The kinetic results indicate that the rate-limiting step for the overall copper leaching process is the diffusion through the ash layer. The optimal conditions for the copper leaching under the range of applied variables are given in Table 10:

## Figures and Tables

**Figure 1 materials-17-00824-f001:**
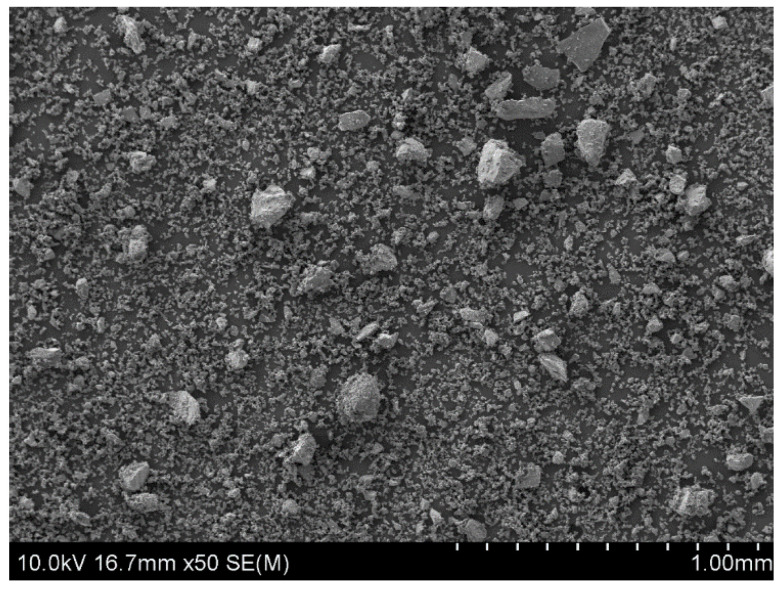
The SEM image of chalcopyrite sample after 5 min grinding in a roller-ring mill. A starting material for the leaching experiments.

**Figure 2 materials-17-00824-f002:**
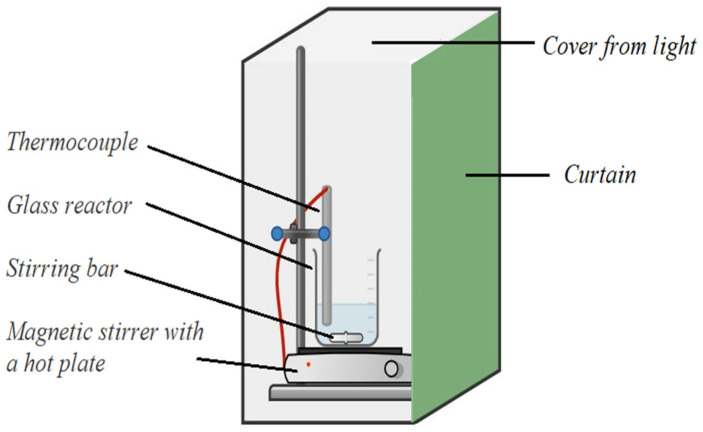
The setup used for experiments of copper leaching from chalcopyrite.

**Figure 3 materials-17-00824-f003:**
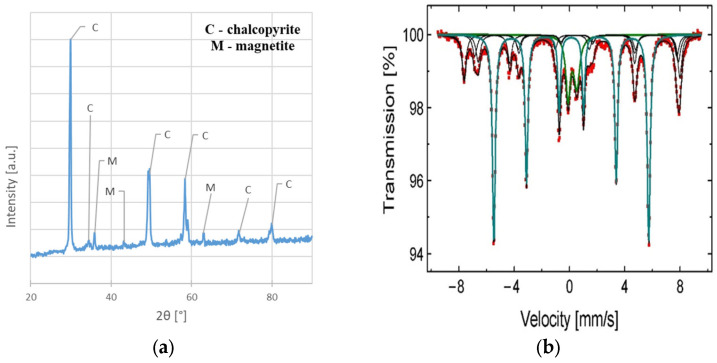
The phase composition analysis of the chalcopyrite sample before the leaching experiments: (**a**) XRD spectrum; (**b**) ^57^Fe Mössbauer spectra characteristic of Fe_3_O_4_, Fe_2_O_3_, and FeS_2_ (obtained at 300 K).

**Figure 4 materials-17-00824-f004:**
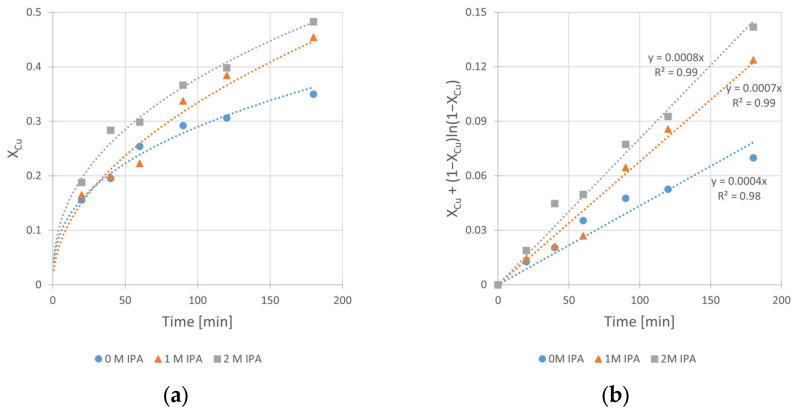
The influence of isopropanol (IPA) concentration on the kinetics of copper leaching from chalcopyrite: (**a**) the conversion (X) over time; (**b**) X_Cu_ + (1 − X_Cu_)ln (1 − X_Cu_) over time. Conditions: T = 40 °C, C_H2SO4_ = 0.5 M, C_H2O2_ = 1 M, v_RPM_ = 400 RPM.

**Figure 5 materials-17-00824-f005:**
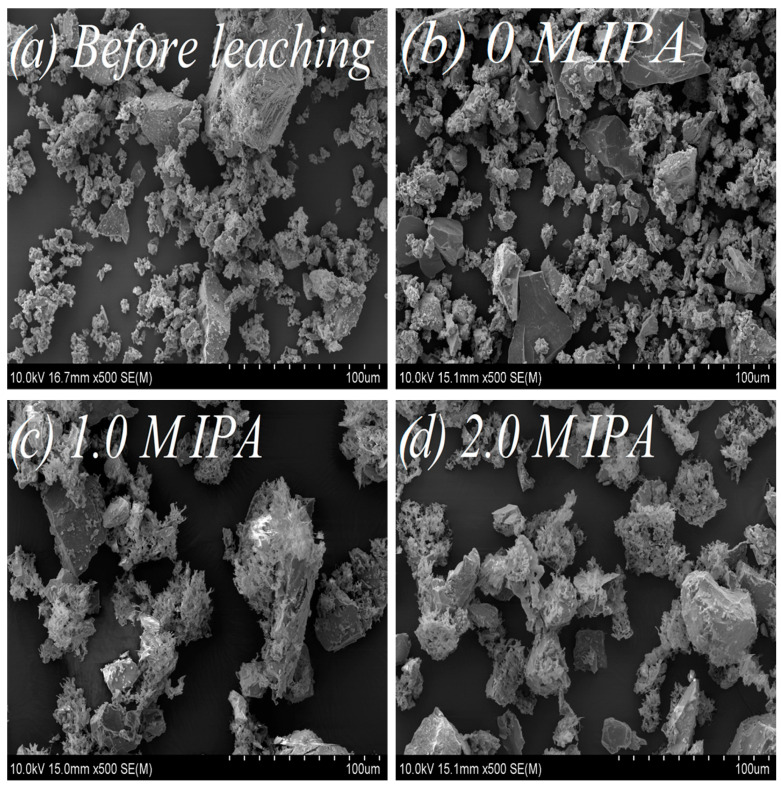
Morphology of the chalcopyrite samples under various leaching conditions: (**a**) before leaching; (**b**) after leaching, without IPA addition (0 M IPA); (**c**) after leaching, in presence of 1.0 M IPA; (**d**) after leaching, in presence of 2.0 M IPA. Magnification: 500–fold.

**Figure 6 materials-17-00824-f006:**
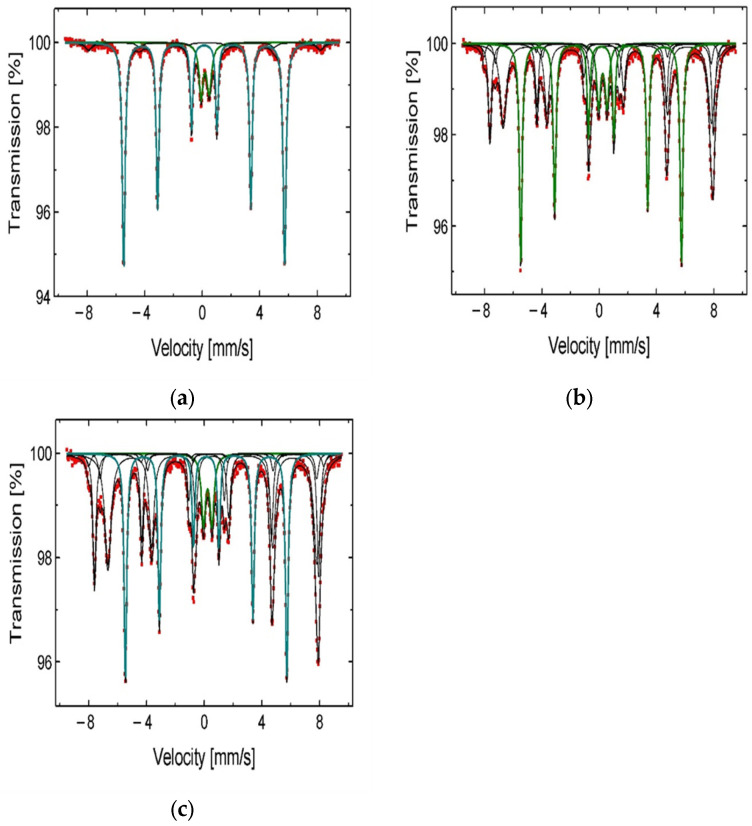
^57^Fe Mössbauer spectra of the chalcopyrite samples after the leaching in H_2_SO_4_ + H_2_O_2_ aqueous solution containing (**a**) 0 M IPA; (**b**) 1.0 M IPA; and (**c**) 2.0 M IPA.

**Figure 7 materials-17-00824-f007:**
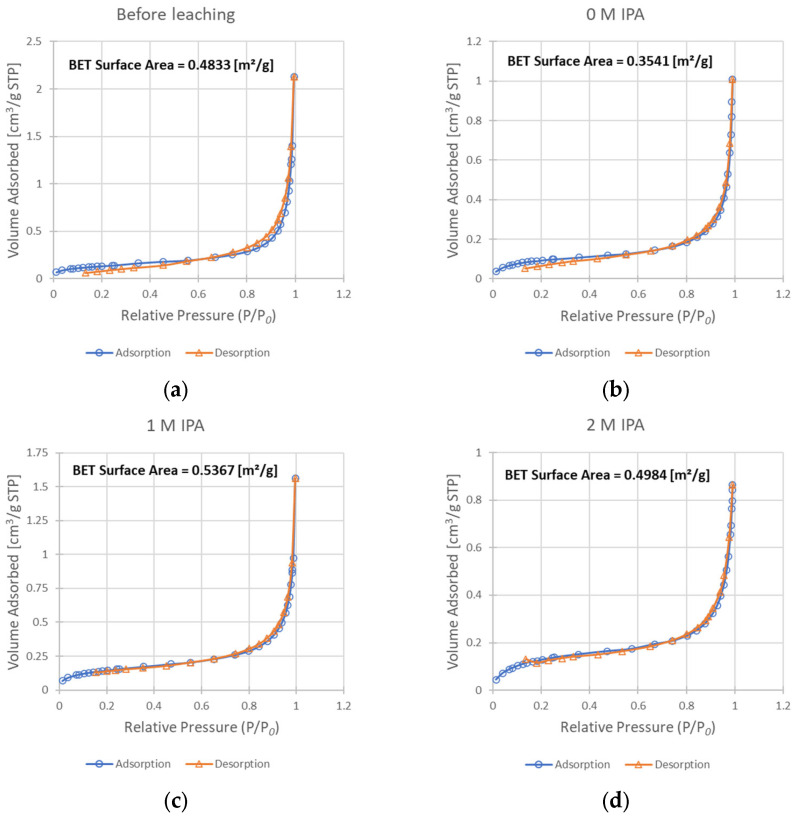
Results of the BET analysis—isotherms of adsorption and desorption, as well as resulting BET surface area values: (**a**) sample before leaching; (**b**) solid residue after leaching without IPA in the system; (**c**) solid residue after leaching with 1 M IPA concentration; (**d**) solid residue after leaching with 2 M IPA.

**Figure 8 materials-17-00824-f008:**
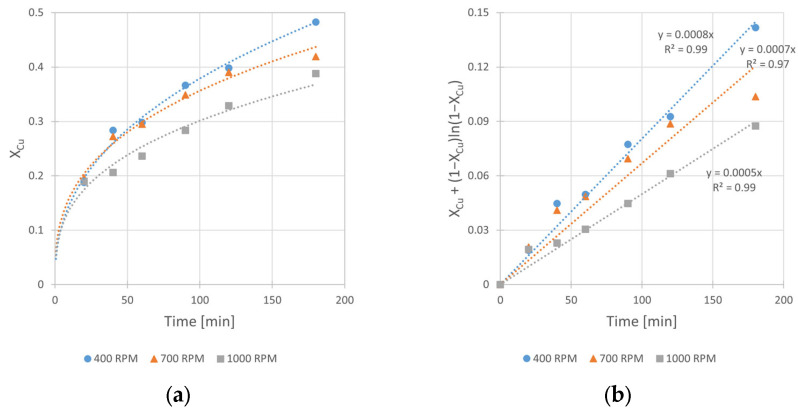
The influence of stirring rate (v_RPM_) on the rate of chalcopyrite leaching: (**a**) the conversion (X) over time; (**b**) X_Cu_ + (1 − X_Cu_)ln (1 − X_Cu_) over time. Conditions: m_Chal._ = 2.0 g, T = 40 °C, V = 200 mL, C_H2O2_ = 1.0 M, C_H2SO4_ = 0.5 M, C_IPA_ = 2.0 M.

**Figure 9 materials-17-00824-f009:**
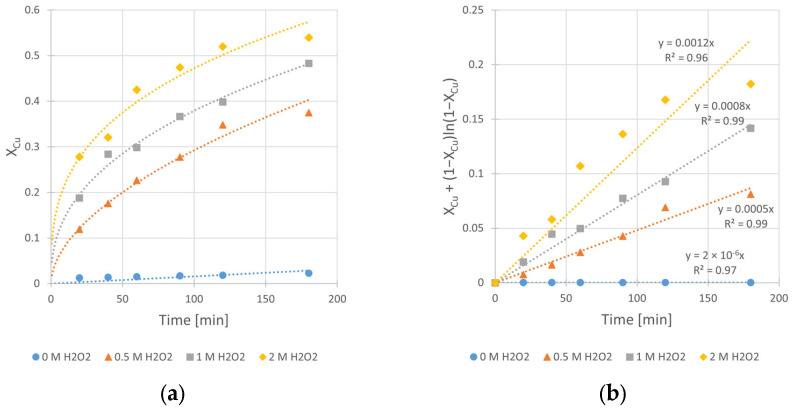
The influence of H_2_O_2_ concentration (C_H2O2_) on the rate of chalcopyrite leaching: (**a**) the conversion (X) over time; (**b**) X_Cu_ + (1 − X_Cu_)ln (1 − X_Cu_) over time. Conditions: m_Chal._ = 2.0 g, T = 40 °C, V = 200 mL, C_H2SO4_ = 0.5 M, C_IPA_ = 2.0 M, *v_SR_* = 400 RPM.

**Figure 10 materials-17-00824-f010:**
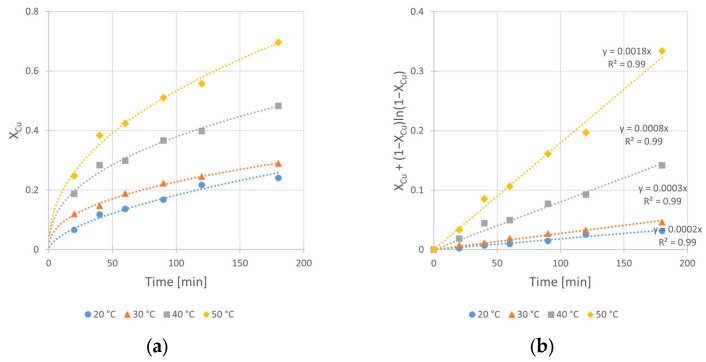
Effect of temperature on the copper leaching kinetics from chalcopyrite: (**a**) conversion (X) over time; (**b**) kinetic curves interpreted in accordance to Shrinking Core Model—Equation (2). Conditions: m_Chal._ = 2.0 g, V = 200 mL, C_H2SO4_ = 0.5 M, C_IPA_ = 2.0 M, C_H2O2_ = 1.0 M, *v_SR_* = 400 RPM.

**Figure 11 materials-17-00824-f011:**
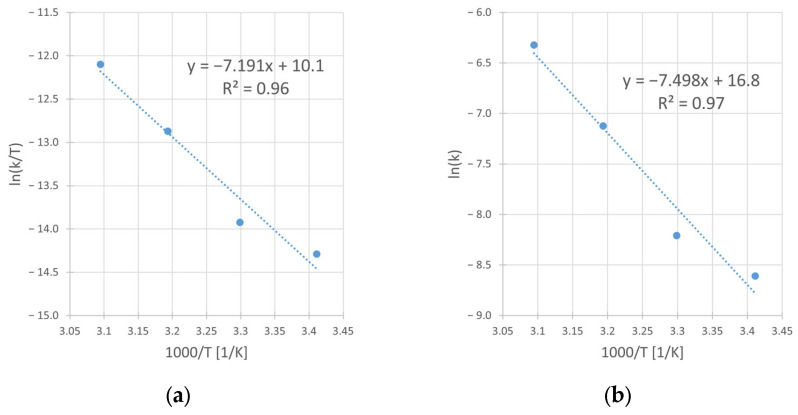
The Eyring (**a**) and Arrhenius (**b**) plots for the reaction of copper leaching from chalcopyrite in the H_2_SO_4_–H_2_O_2_ aqueous medium. Conditions: C_H2SO4_ = 0.5 M, C_IPA_ = 2.0 M, C_H2O2_ = 1.0 M, *v_SR_* = 400 RPM.

**Table 1 materials-17-00824-t001:** The chemical composition of the chalcopyrite (starting material) used in the kinetic experiments—XRF analysis supplemented by IR results.

Element	Cu	Fe	S	Ag	Pb	Zn	Ni	Mn	C
mass. %	31.67	34.55	28.40	2.39	1.50	0.64	0.62	0.22	0.01

**Table 2 materials-17-00824-t002:** Phase identification and corresponding ^57^Fe Mössbauer parameters for sample before leaching (obtained at 300 K).

Phase	Relative Intensity [%]	*IS* [mm/s]	*H* [kGs]	QS [mm/s]	Γ/2 [mm/s]	Reference
Pyrite	12.4	0.318 (5)	0.0	0.314 (4)	0.191 (9)	[20]
Magnetite A-site	14.7	0.297 (5)	483.6 (4)	−0.010 (4)	0.168 (8)	[21]
Magnetite B-site	8.3	0.60 (2)	453.0 (8)	−0.05 (2)	0.189 (9)	[22]
Magnetite B-site	10.5	0.69 (2)	448.5 (7)	0.05 (2)	0.189 (9)	[21]
Chalcopyrite	54.1	0.241 (7)	347.4 (1)	−0.002 (1)	0.129 (2)	[23]

*IS*—isomeric shift; *H*—magnetic hyperfine field; *QS*—quadrupole shift; Γ/2—half of the experimental line width. The numbers in parentheses are the errors on the last digit.

**Table 3 materials-17-00824-t003:** The effect of isopropanol (IPA)—experimental conditions.

Parameter	Value
Chalcopyrite mass (m_Chal._)	2.0 g
Volume of a leaching medium (V)	200 mL
Temperature (T)	40 °C
H_2_SO_4_ concentration (C_H2SO4_)	0.5 M
H_2_O_2_ concentration (C_H2O2_)	1.0 M
Stirring rate (v_SR_)	400 RPM
IPA concentration (C_IPA_)	0, 1.0 and 2.0 M

**Table 4 materials-17-00824-t004:** Phase identification and corresponding ^57^Fe Mössbauer parameters for samples after leaching (obtained at 300 K).

Phase	Relative Intensity [%]	*IS* [mm/s]	*H* [kGs]	QS [mm/s]	Γ/2 [mm/s]	Reference
0 M IPA						
Pyrite	13.3	0.305 (4)	0.0	0.281 (3)	0.164 (8)	[24]
Hematite	5.6	0.35 (3)	500 (2)	−0.08 (3)	0.27 (5)	[25]
Chalcopyrite	81.1	0.242 (1)	347.0 (1)	−0.001 (1)	0.126 (1)	[23]
1 M IPA						
Pyrite	6.8	0.356 (3)	0.0	0.291 (3)	0.185 (7)	[20]
Hematite	2.1	0.36 (1)	507.6 (9)	−0.10 (1)	0.135 (6)	[25]
Magnetite A-site	16.2	0.272 (2)	483.3 (2)	−0.001 (2)	0.135 (6)	[21]
Magnetite B-site	3.2	0.45 (1)	467.1 (9)	−0.06 (1)	0.20 (2)	[26]
Magnetite B-site	28.3	0.668 (4)	450.6 (4)	0.006 (3)	0.255 (9)	[21]
Chalcopyrite	43.4	0.241 (1)	347.6 (1)	−0.002 (1)	0.120 (1)	[23]
2 M IPA						
Pyrite	6.6	0.347 (4)	0.0	0.294 (4)	0.167 (6)	[20]
Hematite	2.3	0.35 (2)	506 (1)	−0.10 (2)	0.167 (6)	[25]
Magnetite A-site	18.7	0.270 (2)	481.7 (2)	−0.003 (2)	0.119 (6)	[21]
Magnetite B-site	3.9	0.47 (2)	466 (1)	−0.07 (1)	0.14 (2)	[26]
Magnetite B-site	32.3	0.668 (4)	449.4 (4)	0.010 (3)	0.247 (9)	[21]
Chalcopyrite	36.2	0.241 (1)	346.9 (5)	−0.001 (1)	0.110 (1)	[23]

*IS*—isomeric shift; *H*—magnetic hyperfine field; *QS*—quadrupole shift; Γ/2—half of the experimental line width. The numbers in parentheses are the errors on the last digit.

**Table 5 materials-17-00824-t005:** The effect of stirring rate—experimental conditions.

Parameter	Value
Chalcopyrite mass (m_Chal._)	2.0 g
Volume of a leaching medium (V)	200 mL
Temperature (T)	40 °C
H_2_SO_4_ concentration (C_H2SO4_)	0.5 M
H_2_O_2_ concentration (C_H2O2_)	1.0 M
IPA concentration (C_IPA_)	2.0 M
Stirring rate (*v_SR_*)	400, 700 and 1000 RPM

**Table 6 materials-17-00824-t006:** The effect of H_2_O_2_ concentration—experimental conditions.

Parameter	Value
Chalcopyrite mass (m_Chal._)	2.0 g
Volume of a leaching medium (V)	200 mL
Temperature (T)	40 °C
H_2_SO_4_ concentration (C_H2SO4_)	0.5 M
IPA concentration (C_IPA_)	2.0 M
Stirring rate (*v_SR_*)	400 RPM
H_2_O_2_ concentration (C_H2O2_)	0, 0.5, 1.0 and 2.0 M

**Table 7 materials-17-00824-t007:** The effect of temperature—experimental conditions.

Parameter	Value
Chalcopyrite mass (m_Chal._)	2.0 g
Volume of a leaching medium (V)	200 mL
H_2_SO_4_ concentration (C_H2SO4_)	0.5 M
H_2_O_2_ concentration (C_H2O2_)	1.0 M
IPA concentration (C_IPA_)	2.0 M
Stirring rate (v_SR_)	400 RPM
Temperature (T)	20, 30 and 40 °C

**Table 8 materials-17-00824-t008:** Enthalpy and entropy of activation for the reaction of copper leaching from chalcopyrite.

Eyring Model			
Slope	Intercept	Enthalpy of activation ∆*H^#^* [kJ/mol]	Entropy of activation ∆*S^#^* [J/mol∙K]
−7191	10.1	59.79	−113.57
Arrhenius Model			
Slope	Intercept	Energy of activation *E_A_* [kJ/mol]	Pre-exponential factor *A* [min^−1^]
−7498	16.8	62.34	1.98∙10^9^

**Table 9 materials-17-00824-t009:** Data comparison taken from different authors of the activation energy (*E_A_*), the leaching medium composition, and SCM model applied for the reaction of copper leaching from chalcopyrite.

Author	Leaching Solution Composition	*E_A_* [kJ/mol]	The Type of SCM
This study	0.5 M H_2_SO_4_ 1 M H_2_O_2_ 2 M IPA	60.68	Cylinder shape, Ash diffusion control
Sokić et al. [12]	1.5 M H_2_SO_4_ 1 M H_2_O_2_	80.00	Flat plate shape, Ash diffusion control
Antonijević et al. [11]	2 M H_2_SO_4_ 2 M H_2_O_2_	60.00	Sphere shape, Reaction control
Solís Marcíal et al. [15]	0.6 M H_2_SO_4_ 3 M H_2_O_2_ 5.7 M CH_3_OH	24.27	Cylinder shape, Reaction control
Adebayo et al. [10]	0.1 M H_2_SO_4_ 5.9 M H_2_O_2_	39.00	Sphere shape, Reaction control

**Table 10 materials-17-00824-t010:** The optimal conditions for efficient copper leaching from the chalcopyrite.

Parameter	Value
Volume of a leaching medium (V)	200 mL
Temperature (T)	50 °C
H_2_SO_4_ concentration (C_H2SO4_)	0.5 M
H_2_O_2_ concentration (C_H2O2_)	1.0 M
IPA concentration (C_IPA_)	2.0 M
Stirring rate (*v_SR_*)	400 RPM

## Data Availability

The datasets used and/or analyzed during the current study are available from T. Michałek (tomaszm@agh.edu.pl) upon reasonable request.

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
