# Peer review of "Chalcopyrite Leaching in the Presence of Isopropanol—The Kinetic and Mechanistic Studies"

_materials, 2024, doi:10.3390/ma17040824_

Round 1
Reviewer 1 Report
Comments and Suggestions for Authors
The contribution is interesting and presents as well important physicochemical information.
Some minor English proofreading should be done to correct some plural/singular misunderstandings, as well as some other fuzzy sentences, e.g. line 155 “Additionally, as a supplementary studies,”, but there are more of this kind, please revise.
In line 47 you state “Both of these phenomena led to the increase of the process efficiency and the kinetics.”, please clarify if there is a “kinetic increase”, perhaps a kinetic increase is not correct, but a “reaction rate increase” should be the term, please rephrase to have a clearer sentence, in its actual state the meaning is fuzzy.
In line 75 you state “(600 ml capacity)”, please use “mL”, and for all the contribution as well (lines 83, 87, etc.).
In lines 75-77 you state “The whole system was protected from an external light source to ensure that semiconductive nature of chalcopyrite will not influence the kinetics.”, please use a reference for these particular conditions.
In section 2. Materials and Methods, therein you lack particular experimental conditions for the obtaining of SEM, XRF, XRD, Mössbauer, and the “C contents” by the IR method, please completely include them in order to reproduce the experimentations. You also must include the software and a concise procedure in which you developed the fitting of the kinetic models employed.
In Figure 3 you should specify which phase corresponds to each spectra in the Mössbauer caption.
In line 144 please correct “4 .10-4 min-1” with “4·10⁻⁴·min⁻¹”.
One important missing in the current contribution is the study of other alcohols in this same kinetic system. In this line, please further revise if other alcohols, perhaps methanol, ethanol, n-propanol, butanols, higher alcohols, etc. have been used for this means, and compare those results with the current. This latter to evidence if the nature of the branched alcohol is a key factor to “protect” H₂O₂ radical decomposition, or if this protection is due to other structural or physicochemical characteristics. You also should determine why the alcohol addition is stabilizing the H₂O₂ species, perhaps not by experiment but by searching references reinforcing this important role in the study, again please further revise this.
Comments on the Quality of English LanguageEnglish could be improved, is not so bad, but some sentences must be improved, as well as some singular/plural grammatical confusions.
Author Response
Thank you for your valuable insights. We believe that the overall quality of our manuscript has increased considerably. A point-by-point response to your comments is included in the attached PDF file.

Reviewer 2 Report
Comments and Suggestions for Authors
The manuscript deals to investigate oxidative copper leaching from chalcopyrite. Beside the kinetic studies, chalcopyrite was characterized by XRD, Mossbauer spectroscopy and SEM. The subject is interesting and up-to date, it is related to the profile of the journal. The structure of the manuscript is generally clear, several up-to date methods were used; however, the method of determination of concluded optimal condition isn’t adequate; it would require more sophisticated statistical analysis. The English of the manuscript is good, few spelling mistakes can be found, please check it carefully.
Comments and questions:
- The Title is informative.
- The Abstract reflects the approach of the study, it summarizes the findings of the work.
- The section Introduction presents the important points of the topic, it contains references related to the earlier results, reveals the importance and originality of the work. However, the mechanism of oxidative leaching should be presented here rather than in the first part of Discussion section.
- Materials and methods: The experimental design is appropriate and generally adequately described.
- How many parallel measurements were performed? Please, indicate the standard deviations of the data.
- Line 87. Please use the correct name. of MP-AES, as Microwave Plasma – Atomic Emission Spectroscopy instead of microwave emission spectroscopy.
- Results and discussion: In this section, authors describe the results shown in the corresponding figures, and tables. The figures generally are nice, and informative, however, the Tables 2, 3 and 4 should contain the varied parameter too.
- Line 114. Please correct indices.
- Section 3.2. Did you examine lower stirring rates? If not, it cannot be concluded that 400 rpm was the optimal; it was the best among the investigated stirring rates. For seeking optimal conditions, design of experiments is a more sophisticated method.
- Discussion section discuss the results in detailed and explain the mechanism well.
- The Conclusions section summarizes shortly the result of the work.
- Conclusion and recommendation: This manuscript is recommended for publication after minor revision.
Author Response

(The authors gave the same response as above.)

Reviewer 3 Report
Comments and Suggestions for Authors
In this paper, the effects of IPA, H2O2 concentration, stirring rate and temperature on the leaching rate and efficiency of copper from chalcopyrite were studied. The kinetic curves obtained by the experiment were analyzed using the SCM. The positive effects of IPA and H2O2 concentration and temperature on copper leaching kinetics and efficiency were confirmed by the copper leaching rate constant (kobs). Finally, the optimum conditions of effective leaching, the speed limiting steps and the mechanism of copper dissolution are given. However, there are still some problems that need to be solved.
1. “Page 3”
“Table 1. The chemical composition of the chalcopyrite (starting material) used in the kinetic exper-iments–XRF analysis supplemented by IR results”
The sum of each chemical composition exceeds 100%, please check.
2. “Page 4”
“The results also indicated a greater Fe and S content than would appear from the CuFeS2 stoi- chiometry.”
Please elaborate on how this conclusion was reached.
3.“Page 4”
“Figure 3. The phase composition analysis of the chalcopyrite sample before the leaching experi- ments: a) XRD spectrum; b) Mossbauer spectra characteristic of Fe3O4, Fe2O3 and FeS2 (obtained at 300 K). ”
Please explain what each color symbol in Figure 3 (b) represents.
4.“Page 5”
“However, due to the fact that 2.0 M value has proven to be the most effective one, further kinetic experiments were performed at this concentration value.”
In experiments with IPA content of 0 M, 1.0 M, 2.0 M, the content of 2.0 M is the relatively effective concentration. Please explain how you came to the conclusion that 2.0M has proven to be the most effective concentration.
5.“Page 5”“Page 7”
“Additionally, as a supplementary studies, the change of the substrate (chalcopyrite) surface was estimated from the measurements of N2 adsorption, using BET isotherm model.”and“The comparison of BET iso- therm data for the un-leached sample and the leached one (at 2 M IPA presence) showed the increase of the surface area from 0.511 (+/- 0.006) to 0.677 (+/- 0.016) m2/g.”
Please provide relevant experimental data to supplement the research content.
6.“Page 7”
“The analysis of experimental kinetic curves (Figure 6) shows that increasing rate of stirring has a negative impact on the leaching rate.”
Please check if it is Figure 6.
7. “Page 10”
“From the data shown in Figure 8 the effect of increasing temperature from 20 to 50°C is evident.”
Please check if the description is Figure 8.
8. “Page 12”
“It can be also deduced from the data shown in Table 7 that increasing H2O2 concentration in leaching medium one can decrease the EA value.”
Not controlling for a single variable, this conclusion cannot be drawn from Table 7.
Comments on the Quality of English LanguageMinor editing of English language required
Author Response

(The authors gave the same response as above.)
